# Improving Semantic Parsing with Neural Generator-Reranker Architecture

## Abstract

Semantic parsing is the problem of deriving machine interpretable meaning representations from natural language utterances. Neural models with encoder-decoder architectures have recently achieved substantial improvements over traditional methods. Although neural semantic parsers appear to have relatively high recall using large beam sizes, there is room for improvement with respect to one-best precision. In this work, we propose a generator-reranker architecture for semantic parsing. The generator produces a list of potential candidates and the reranker, which consists of a pre-processing step for the candidates followed by a novel critic network, reranks these candidates based on the similarity between each candidate and the input sentence. We show the advantages of this approach along with how it improves the parsing performance through extensive analysis. We experiment our model on three semantic parsing datasets (GEO, ATIS, and OVERNIGHT). The overall architecture achieves the state-of-the-art results in all three datasets.

## 1 Introduction

Semantic parsing is the task of deriving machine interpretable meaning representations such as logical forms or structured queries from natural language utterances. These meaning representations can be executed in various environments, making semantic parsing applicable in many frameworks such as querying data/knowledge bases for question answering (Zelle & Mooney, 1996; Zettlemoyer & Collins, 2007; Liang et al., 2011; Berant et al., 2013), generating regular expression (Kushman & Barzilay, 2013), instruction following (Artzi & Zettlemoyer, 2013), and communicating with robots (Chen & Mooney, 2011; Tellex et al., 2011; Bisk et al., 2016).

Conventionally, semantic parsing has been done with a two step approach: first, a large number of potential candidates are generated using deterministic methods and combinatorial search (the generator), and then the best candidate is selected among them with a probabilistic method or scoring (the critic) (Berant & Liang, 2014; Kwiatkowski et al., 2013; Zettlemoyer & Collins, 2005; Berant et al., 2013; Cai & Yates, 2013). With recent advancement of neural networks, neural models with encoder-decoder architectures has obtained impressive improvements (Dong & Lapata, 2016; Jia & Liang, 2016; Herzig & Berant, 2017; Su & Yan, 2017; B. Chen & Han, 2018; Shaw et al., 2019). These encoder-decoder based neural semantic parsers produce one candidate given an input sentence, essentially acting as both the generator and the critic. However, undesirable prediction errors still occur, e.g. predicting a wrong comparative or superlative structure (such as $<$ instead of $\leq$). On the other hand, neural semantic parsing models have a high recall, i.e. top-$n$ predictions of the model cover the gold-standard meaning representation most of the time (c.f. Table 2 in Section 2).

In this work we propose a generator-reranker architecture that uses two neural networks for semantic parsing: a generator network, which generates a list of potential candidates, and a reranker system, which consists of a pre-processing step for the candidates followed by a novel critic network that reranks these candidates based on the similarity between each candidate and the input sentence. An advantage of separating the semantic parsing process into a generator network and a critic network is that the critic observes each candidate and the input sentence entirely, taking into account bidirectional representations of both sentences and can globally reason over the entire candidate. This may be more effective in terms of choosing the right candidate and mitigating some of the errors arising from auto-regressive decoding in the generator. Furthermore, the critic can leverage extra

Table 1: One example from each dataset that we use in our experiments. Input utterances and the corresponding logical forms are denoted by $x$ and $y$ respectively.

| Dataset | Example |
|---|---|
| GEO | $x$ : *"which states adjoin alabama ?"* |
| | $y$ : `answer(state(next_to_2(stateid(alabama))))` |
| ATIS | $x$ : *"get flights between st. petersburg and charlotte"* |
| | $y$ : `(_lambda $0 e (_and (_flight $0)` |
| | `(_from $0 st_petersburg:_ci) (_to $0 charlotte:_ci)))` |
| OVERNIGHT | $x$ : *"show me all meetings not ending at 10 am"* |
| | $y$ : `Type.Meeting ⊓ EndTime.  != 10` |

data sources for training, e.g. a paraphrase dataset and allow for better transfer learning. Our key contributions in this work are the following:

1. We introduce a neural critic model, which reranks the candidates of a semantic parser based on their similarity score with respect to the input utterance.

2. We propose various pre-processing methods for the candidates to leverage the pre-trained representations of their tokens using the critic model.

3. We show through extensive qualitative and quantitative studies how the critic model helps mitigating errors of a state-of-the-art neural semantic parser and improves the performance.

Evaluation results of our approach on three existing semantic parsing datasets (see Table 1 for a sample input-output pair for each dataset) show that our model improves upon the state-of-the-art results and the generator-reranker architecture can substantially improve parsing performance.

## 2 RELATED WORK

The semantic parsing problem has received significant attention and has a rich literature (Kamath & Das, 2018). While traditional approaches (Kate & Mooney, 2006; Wong & Mooney, 2007; Clarke et al., 2010; Zettlemoyer & Collins, 2007; Kwiatkowski et al., 2011; Wang et al., 2015; Li et al., 2015; Cai & Yates, 2013; Berant et al., 2013; Quirk et al., 2015; Artzi et al., 2015; Zhang et al., 2017) rely on high-quality lexicons, manually-built templates, and/or domain or representation specific features, in more recent studies neural models with encoder-decoder architectures show impressing results (Dong & Lapata, 2016; Jia & Liang, 2016; Herzig & Berant, 2017; Su & Yan, 2017; B. Chen & Han, 2018; Shaw et al., 2019).

Among recent results, Herzig & Berant (2017) achieves the best performance on the OVERNIGHT dataset (Wang et al., 2015), which consists of 8 various domains. The model in Herzig & Berant (2017) is an attention-based sequence-to-sequence model and it is trained jointly over 8 domains. Very recently, Shaw et al. (2019) presents an approach that uses a Graph Neural Network (GNN) architecture, which successfully incorporates information about relevant entities and their relations in parsing natural utterances. Similar to Vinyals et al. (2015); Jia & Liang (2016); Herzig & Berant (2017), the decoder has a copying mechanism, which can copy an entity to the output during parsing. This model achieves the best performance in GEO dataset (Zelle & Mooney, 1996) and is competitive with state-of-the-art in ATIS dataset (Dahl et al., 1994). We note that this model was not applied to the OVERNIGHT dataset.

We provide in Table 2 the comparison between top-10 (25) oracle[1] and top-1 (greedy decoding) accuracy for the state-of-the-art sequence-to-sequence models. We note that while it may not be possible to reach top-10 (25) accuracy for all datasets[2], there is certainly room for improvement.

---

[1] The fraction of test examples where the gold-standard logical form is present in one of the top-10 (25) predicted logical forms.

[2] For instance, Wang et al. (2015) reports 17% rate of incorrect labels in the OVERNIGHT dataset.

Table 2: Top-1 (greedy decoding) vs top-10 (25) oracle for the best performing sequence-to-sequence models on three semantic parsing datasets.

| Model | Dataset | Top-1 Acc. | Top-10 Oracle | Top-25 Oracle |
|---|---|---|---|---|
| Herzig & Berant (2017) | OVERNIGHT | 79.6 | 91.7 | 93.5 |
| Shaw et al. (2019) | GEO | 92.5 | 95.3 | 96.4 |
| Shaw et al. (2019) | ATIS | 89.7 | 93.3 | 94.2 |

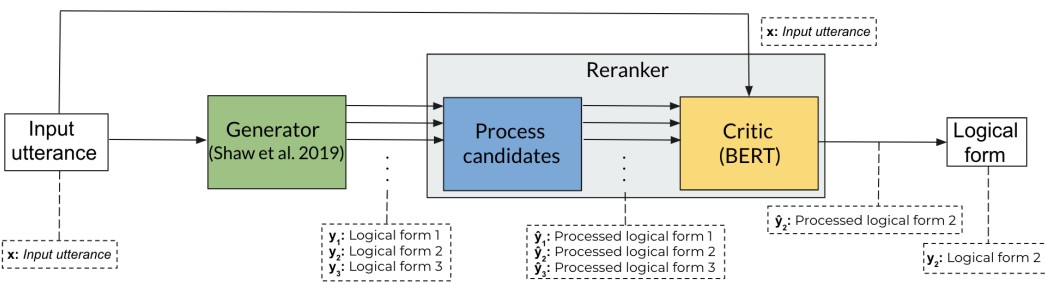

Figure 1: Overview of our method demonstrating semantic parsing process. In the example, the top prediction by the generator is $y_1$ whereas the candidate $y_2$ is scored highest by the critic among the candidates of the generator.

Motivated by these observations, we consider a generator-reranker architecture for semantic parsing. The reranker system consists of a critic network, which reranks candidate logical forms based on their similarity scores to the input utterance. We propose various techniques for processing candidate logical forms before the critic scores their similarity to the input utterance. For the generator, we use the model in Shaw et al. (2019) and for the critic, we use the BERT model (Devlin et al., 2018) for all three datasets. Although the setting is parser agnostic, we would like to investigate if the generator-reranker architecture can further improve the performance of the best performing parser.

To the best of our knowledge, our reranker system is novel and has not been considered before in semantic parsing. However, the idea of reranking the candidates of a generator model appears in various applications in the literature. We refer the reader to Collins & Koo (2005) for reranking based on a combination of feature functions for various NLP tasks.

For semantic parsing task, Yavuz et al. (2016) applies reranking based on predicted answer type. We note that this method may not be effective for the cases where the candidate logical forms return the same answer type but the top prediction is wrong (such as predicting a wrong comparative, e.g. $<$ instead of $\leq$). We believe choosing the best candidate by assessing its relevance to the input utterance entirely would be a more effective method. In a more recent work by Yin & Neubig (2019), reranking is applied by two main quality-measuring features of candidate logical forms. The first one is the reconstruction feature, using the probability of reproducing the original input utterance $x$ from $y$. The second is the discriminative matching feature, which is based on pair-wise associations of tokens in $x$ and $y$. In our work, reranking of candidate logical forms is applied based on their similarity with the input utterance directly using a critic model which can leverage the pre-trained representations of processed logical forms as well as extra data sources to learn similarity.

## 3 MODEL ARCHITECTURE

Our goal is to learn a model which maps an input utterance $x$ to a logical form representation of its meaning $y$. The input utterance $x$ is a sequence of words $x_1, x_2, \ldots, x_{n_1} \in \mathcal{V}^{(\text{in})}$ where $\mathcal{V}^{(\text{in})}$ is the input vocabulary and the output logical form $y$ is a sequence of tokens $y_1, y_2, \ldots, y_{n_2} \in \mathcal{V}^{(\text{out})}$ where $\mathcal{V}^{(\text{out})}$ is the output vocabulary.

In this work, we consider a combination of two models for semantic parsing. Figure 1 illustrates the model architecture along with an example. We describe the model architecture in what follows.

## 3.1 GENERATOR

To generate candidate logical forms, we use the model recently introduced in Shaw et al. (2019), which is based on the Transformer architecture (Vaswani et al., 2017), with the self-attention layer extended to incorporate relations between input elements, and the decoder extended with a copy mechanism similar to Vinyals et al. (2015); Jia & Liang (2016); Herzig & Berant (2017). The model is trained with natural utterances paired with logical forms and learns shared representations from these pairs. Candidate logical forms are generated using beam search for a given input utterance.

## 3.2 PROCESSING CANDIDATES

Before scoring the similarity between a logical form and an input utterance, we preprocess the candidate logical forms. We propose the processing methods in an increasing order of their complexity and how close they map the logical forms to natural text. Note that the critic can leverage the pretrained representations of more processed logical forms better and can be more effective when scoring similarities. Figure 3 in Appendix A illustrates these methods along with an example.

### 3.2.1 RAW LOGICAL FORM

In this method, we consider raw logical forms without any processing for calculating similarity with respect to the input utterance. This is the simplest method that is applicable to any dataset.

### 3.2.2 NATURAL LANGUAGE ENTITY NAMES

In this method, we simply convert entities to natural text. The output tokens are often self-explanatory and easy to be converted to simple text, e.g., "*num_assists*" → "number of assists", "*en.location.greenberg_cafe*" → "greenberg cafe" in OVERNIGHT dataset, and "*_arrival_time*" → "arrival time" in ATIS dataset etc. This approach requires an additional step but can be applied in a straightforward manner. The advantage of this approach is that the critic can leverage pre-trained representations for these tokens. We describe the exact procedure in Appendix A for each dataset.

### 3.2.3 TEMPLATED EXPANSIONS

In this method, logical forms are converted to canonical utterances using a deterministic template (e.g. `arg max(type.player, numRebounds)` to "player that has the largest number of rebounds"). The purpose of the canonical utterances is to capture the meaning of the logical forms. Here the assumption is that while it is nearly impossible to generate a grammar that parses all utterances, it is possible to write one that generates one canonical utterance for each logical form (Wang et al., 2015). We use this method only for OVERNIGHT dataset as there is an available template introduced in Wang et al. (2015) to convert the logical forms to canonical utterances. In this case, the critic can be considered as a paraphrase model, producing a score based on the equivalence of two sentences semantically. During test time, candidate logical forms that do not have a corresponding canonical utterance are treated as incorrect candidates and they are not included in ranking.

## 3.3 CRITIC

The critic network takes two sentences as an input and outputs a score $s \in [0, 1]$ based on the similarity of the sentences. The input may consist of two arbitrary sentences (coming from different language models, vocabularies etc.) depending on the processing of the logical forms described in the previous section. We use the critic to rerank the candidate logical forms based on their similarity score with respect to the input utterance. We use the BERT model (Devlin et al., 2018) for this task[3].

To train the critic, positive and negative examples are generated as follows. Each natural utterance in the training set has a gold-standard logical form, which forms the positive examples. For the negative examples we apply the generator, which is trained over the training set, to generate logical forms for each natural utterance in the training set with beam search. The incorrect logical forms are paired with the natural utterance for negative examples. Additionally, we pair any two logical forms

---

[3]The logical forms are tokenized using the public BERT wordpiece tokenizer as wordpieces often contain useful information. Different tokenization methods can be considered, which we leave for future work.

Figure 2: Overview of generating training examples for the critic.

among beam candidates as negative examples to increase training data and to help the model learn subtle differences among the candidates. The logical forms are processed according to the method we use in the model. The positive pairs are labeled as 1 and the negative pairs are labeled as 0. The model performs binary classification. Figure 2 illustrates this approach with an example.

As aforementioned, we can leverage existing paraphrase datasets to pretrain the critic and fine tune it over the examples we generate. For instance, we use the Quora question pairs[4], which contains over 400K annotated question pairs with binary paraphrase labels. Furthermore, we may choose not to apply reranking in certain cases. From the error cases of the critic over the training examples, two cases stand out. The first one is where each candidate is scored below 0.5 (no candidate is similar to the input utterance according to the critic) and the second one is the case where at least two candidates are scored very high and close to each other. It appears quite natural not to do ranking if all scores are below 0.5 to address the first case. On the other hand, one can address the second case by setting a threshold between the best and second best score and choose not to do ranking if the difference is less than this threshold. However, setting this threshold is prone to overfitting and may look somewhat arbitrary if chosen by maximizing the accuracy over the evaluation set. Therefore, we set this threshold as 0.001 once and keep it same for all experiments. Our goal here is merely to show various strategies one can take with the critic model. We may choose not to rerank if either or both of these cases occur at inference and instead output the top prediction of the generator.

## 4 EXPERIMENTS

### 4.1 DATASET

We use three semantic parsing datasets in our experiments. Table 1 presents an example for each dataset.

**GeoQuery** (`GEO`) contains natural language questions about US geography along with corresponding logical forms (Zelle & Mooney, 1996). We follow Zettlemoyer & Collins (2005) and use 600 training examples and 280 test examples. We use logical forms based on Functional Query Language (FunQL) (Kate et al., 2005). We use logical form exact match when reporting accuracy.

**ATIS** (`ATIS`) contains natural language queries for a flights database along with corresponding database queries written in lambda calculus (Dahl et al., 1994). We follow Zettlemoyer & Collins (2007) and use 4473 training examples and 448 test examples. We compare normalized logical forms using canonical variable naming and sorting for unordered arguments (Jia & Liang, 2016) when reporing accuracy.

**Overnight** (`OVERNIGHT`) contains 13,682 examples of language utterances paired with logical forms across eight domains (Wang et al., 2015). In this dataset, each logical form has a corresponding canonical utterance, which we use in templated expansions method when we process the candidate logical forms to canonical utterances. We evaluate on the same train/test split as Wang et al. (2015); Jia & Liang (2016); Herzig & Berant (2017) with the same accuracy metric, i.e. the fraction of test examples for which the denotations of the predicted and gold logical forms are equal.

### 4.2 TRAINING DETAILS

For the generator, we follow the settings of Shaw et al. (2019) for `GEO` and `ATIS` datasets. The model was not applied to `OVERNIGHT` dataset, therefore, we configured the hyperparameters based

---

[4]See https://data.quora.com/First-Quora-Dataset-Release-Question-Pairs.

Table 3: Test accuracy for all models on `OVERNIGHT` dataset, which has eight domains: Basketball, Blocks, Calendar, Housing, Publications, Recipes, Restaurants, and Social. We use the generator-reranker (G-R) architecture with different options. Beam-$n$: Beam search is applied with size $n$, pQ: The critic is pre-trained over the Quora dataset, TH1: rerank if there is at least one score above 0.5, TH2: rerank if best score $-$ second best score $> 0.001$. The candidate logical forms are processed with templated expansions method (Section 3.2.3) in this experiment.

| Method | Bas. | Blo. | Cal. | Hou. | Pub. | Rec. | Res. | Soc. | Avg. |
|---|---|---|---|---|---|---|---|---|---|
| **Previous Methods** | | | | | | | | | |
| B. Chen & Han (2018) | 88.2 | 61.4 | 81.5 | 74.1 | 80.7 | 82.9 | 80.7 | 82.1 | 79.0 |
| Su & Yan (2017)[5] | 88.2 | 62.2 | 82.1 | 78.8 | 80.1 | 86.1 | 83.7 | 83.1 | 80.6 |
| Herzig & Berant (2017) | 86.2 | 62.7 | 82.1 | 78.3 | 80.7 | 82.9 | 82.2 | 81.7 | 79.6 |
| **Our Methods** | | | | | | | | | |
| Shaw et al. (2019) | **89.3** | 63.7 | 81.5 | 82.0 | 80.7 | 85.6 | 89.5 | 84.8 | 82.1 |
| G-R (Beam-10) | 88.7 | 66.4 | 83.3 | 82.5 | 78.9 | 86.6 | 89.8 | 83.7 | 82.5 |
| G-R (Beam-10 & pQ) | 89.0 | 65.2 | 83.3 | 83.6 | 78.3 | 87.5 | 89.5 | 85.5 | 82.7 |
| G-R (Beam-25) | 89.0 | **67.7** | 83.3 | **84.1** | **82.6** | 87.5 | 89.4 | 83.9 | 83.4 |
| G-R (Beam-25 & pQ) | **89.3** | 66.7 | 84.5 | 83.6 | 80.1 | **88.0** | **91.0** | 85.2 | 83.5 |
| G-R (Beam-25 & pQ & TH1) | 89.0 | 65.7 | **85.1** | 83.6 | 81.4 | **88.0** | **91.0** | **86.0** | **83.7** |
| G-R (Beam-25 & pQ & TH2) | 88.7 | 66.4 | 82.7 | 83.1 | 82.0 | 87.0 | 89.8 | 85.8 | 83.2 |

on performance cross-validated on the training set. We provide the final setting and training process in full detail in Appendix B.

For the critic, we use the BERT$_{LARGE}$ model in Devlin et al. (2018), where we feed two sentences as an input and feed the `[CLS]` representation into an output layer for binary classification. We either directly train the model with the examples generated by the generator (see Section 3.3) or train the model on Quora question pairs first and fine tune it over our examples. We apply the early stopping rule based on the evaluation set accuracy to determine the total training steps. We use the learning rate 1e-6 and batch size 32 when we train and fine tune the critic.

## 4.3 RESULTS

**OVERNIGHT:** We compare our model with the state-of-the-art models in Table 3. All models use the same training/test splits, therefore, we directly take the reported best performances from their original papers for fair comparison. We use the model in Shaw et al. (2019) for the generator. As Table 3 shows, this model alone (without any reranking) improves the state-of-the-art performance from 79.6% to 82.15% accuracy and sets a new state-of-the-art as a sequence-to-sequence model. In this experiment, we use templated expansions method (Section 3.2.3) when processing candidate logical forms for reranking. We do our experiments over the generator-reranker architecture on three settings:

1. **Beam size** - This determines the number of candidates produced by the generator.
2. **Initialization of the critic** - This determines how the critic is initialized. We use a pre-trained model (pQ), which is trained over Quora question pairs and fine tuned over the generated examples by the generator. We compare this with directly training the critic over the generated examples by the generator without Quora dataset.
3. **Reranking with a threshold rule** - We apply reranking based on various threshold rules. We compare reranking at all times with reranking when there is at least one candidate with score above 0.5 (TH1) and reranking if the difference between best score and second best score is above 0.001 (TH2).

From the overall results, we can see that:

---

[5]In Su & Yan (2017), each domain has a separate model where it is trained using the training data of all the other domains and fine tuned on the training data of the domain reported. All the other methods have one single model.

1. Increasing the beam size improves the performance as expected. As the generator outputs more candidates, it is more likely that the correct form is among them. Therefore, this allows the critic to identify a higher number of true positives and improve the performance. Increasing the beam size further does not significantly improve the performance (beam size 50 achieves 83.80% accuracy), hence we conclude the experiment with beam size 25.

2. Using a pre-trained model improves the performance as well. We note that the task of the critic is to infer the similarity of two input sentences, therefore, we can initialize it with one that has been trained over an existing paraphrase dataset.

3. Reranking with a threshold rule may be helpful for the overall architecture. We observe that reranking by the critic at all times may not be the best approach. We note that choosing not to rerank when all scores are below 0.5 increases the performance further. On the other hand, reranking if the difference between the best score and second best score is above the threshold we set does not help in this case.

The overall architecture improves the performance of the generator (82.1% accuracy) to 83.7% accuracy. We note that this improvement is significant for `OVERNIGHT` dataset as it is an average over 8 domains with an improvement for each one of them.

We next apply raw logical form (Section 3.2.1) and natural language entity names (Section 3.2.2) methods when processing the candidate logical forms and show that the critic improves the performance in these cases as well. While our best result with templated expansions (Section 3.2.3) method achieves 83.71% accuracy in Table 3, the best results we achieve with the first two methods are 82.81% and 83.16% respectively. We note that the critic improves upon the performance of the generator (82.1% accuracy) in all three cases. As the logical forms are processed more towards natural text, the performance gets better. This is expected since it helps the critic to measure the similarity with respect to the input utterance.

**GEO and ATIS:**  We continue with `GEO` and `ATIS` datasets and provide the best set of our results.

For the `GEO` dataset, we set the beam size as 10 for the number of candidates produced by the generator. We do not process the candidate logical forms and use the raw versions as the output tokens are already in processed form. We pretrain the critic over the Quora dataset and fine tune it over the generated examples by the generator on the training set. We produce 25 candidate logical forms for each example in the training set when generating examples to train the critic.

For the `ATIS` dataset, we set the beam size as 10 for the number of candidates produced by the generator. We process the candidate logical forms with natural language entity names method in a straightforward manner (see Appendix A.1). We pretrain the critic over the Quora dataset and fine tune it over the generated examples by the generator on the training set. We produce 25 candidate logical forms for each example in the training set when generating examples to train the critic.

The results are shown in Table 4. We observe a performance gain in both datasets and achieve the state-of-the-art performance with the generator-reranker architecture. We note that in `ATIS` dataset, there is a significant improvement upon the baseline and the overall architecture obtains the state-of-the-art result over Wang et al. (2014), where the approach is not based on neural models.

## 4.4 ERROR ANALYSIS

In this section, we categorize the types of errors the generator model makes and analyze which ones are corrected by the critic to understand what types of errors can be mitigated via our approach. We use the `OVERNIGHT` dataset as it has a much larger test set. In the following examples, the top prediction of the generator is wrong, but the gold-standard logical form is among the beam candidates and correctly scored highest by the critic. We provide the corresponding canonical utterances instead of the logical forms for the sake of presentation.

1. **Wrong comparative structure:**
   *Input utterance:* meetings after january 2 or after january 3
   *Top prediction:* meeting whose date is at least jan 2 or jan 3
   *Corrected to:* meeting whose date is larger than jan 2 or jan 3

Table 4: Test accuracy for all models on `GEO` and `ATIS` datasets. The settings follow the same as Table 3 and we denote TH3 as the threshold rule of applying both TH1 and TH2, i.e. reranking when there is at least one score above 0.5 and if best score − second best score > 0.001.

| Method | GEO |
|---|---|
| **Previous Methods** | |
| Shaw et al. (2019) | 92.5 |
| **Our Methods** | |
| G-R (Beam-10 & pQ) | 92.5 |
| G-R (Beam-10 & pQ & TH1) | 92.5 |
| G-R (Beam-10 & pQ & TH2) | **93.2** |
| G-R (Beam-10 & pQ & TH3) | **93.2** |

| Method | ATIS |
|---|---|
| **Previous Methods** | |
| Wang et al. (2014) | 91.3 |
| Shaw et al. (2019) | 89.7 |
| **Our Methods** | |
| G-R (Beam-10 & pQ) | 90.6 |
| G-R (Beam-10 & pQ & TH1) | 91.1 |
| G-R (Beam-10 & pQ & TH2) | 91.3 |
| G-R (Beam-10 & pQ & TH3) | **91.5** |

2. **NP-shift to wrong position**:
   *Input utterance:* what article citing multivariate data analysis was in annals of statistics
   *Top prediction:* article whose venue is annals of statistics and that multivariate data analysis cites
   *Corrected to:* article whose venue is annals of statistics and that cites multivariate data analysis

3. **Similar word confusion in elliptical constructions**:
   *Input utterance:* show me recipes with a preparation time equal to or greater than rice pudding
   *Top prediction:* recipe whose preparation time is at least cooking time of rice pudding
   *Corrected to:* recipe whose preparation time is at least preparation time of rice pudding

4. **Incorrect matching to semantically non-equivalent phrase**:
   *Input utterance:* housing that is cheaper than 123 sesame street
   *Top prediction:* housing unit whose size is smaller than size of 123 sesame street
   *Corrected to:* housing unit whose monthly rent is smaller than monthly rent of 123 sesame street

Although the critic helps fixing these errors and improves the performance in semantic parsing, there are few cases where it is ineffective. Therefore, a stronger critic may further increase the performance. We provide here the cases where the top prediction of the generator is correct, but the critic scores another prediction higher and this leads to an error.

1. **Comparative by trivial algebra:**
   *Input utterance:* friends of people who joined their jobs before 2005
   *Top prediction:* person that employee whose start date is at most 2004 is friends with
   *Best scored candidate:* person that employee whose start date is smaller than 2004 is friends with

2. **Comparative by common sense:**
   *Input utterance:* who is younger than alice
   *Top prediction:* person whose birthdate is larger than birthdate of alice
   *Best scored candidate:* person whose birthdate is smaller than birthdate of alice

# 5 CONCLUSION

In this paper, we proposed a generator-reranker architecture for semantic parsing. We introduced a novel neural critic model that reranks the candidates of a semantic parser based on their similarity scores with respect to the input utterance. We proposed various processing methods for the candidate logical forms, enabling the critic to leverage the pre-trained representations for the tokens of the candidates. Our architecture further enables leveraging extra data resources in a direct fashion. We showed that the proposed architecture improves the parsing performance and achieves the state-of-the-art results on three semantic parsing datasets.

As the model is generator agnostic, in the future work we plan to try (a combination of) neural and/or traditional parsers as the generator and apply our architecture on more benchmark datasets. In addition, we believe that our architecture may also be effective in the cross-lingual semantic parsing setting (Zhang et al., 2018). One can leverage paraphrase datasets available in source-target language and pretrain the critic, which could help choosing the right form among the generator candidates.

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

## A  PROCESSING CANDIDATES

### A.1  NATURAL LANGUAGE ENTITY NAMES

We apply this method to OVERNIGHT and ATIS datasets as the GEO dataset does not require processing since the output tokens already has English words.

For the ATIS dataset, we apply a very simple procedure. We take the tokens that start with an underscore (i.e. "_") and remove it (e.g. "_from" → "from", "_to" → "to" etc.). Additionally, for such tokens, if there are underscores in between, we replace them with space to break into further tokens (e.g. "_departure_time" → "departure time", "_fare_amount" → "fare amount").

For the OVERNIGHT dataset, each entity token has a corresponding English text available in Wang et al. (2015), which is used in the process of building the dataset. These are used within a set of rules for the logical form, canonical utterance conversion as well. We note that this method is rather straightforward compared to templated expansions method since we only replace each entity token with its corresponding English text.

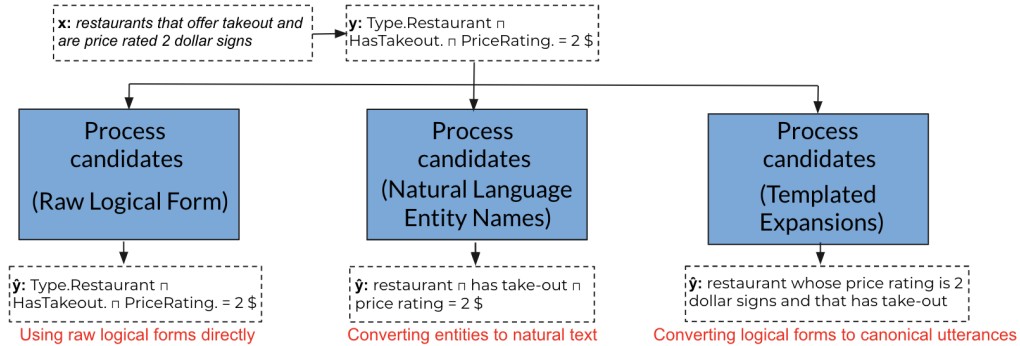

Figure 3: Overview of the proposed three methods with an example demonstrating each one. Raw logical form method uses logical forms as they are without any change, natural language entity names method simply changes entities to natural text, and templated expansions method uses a template to convert logical forms to canonical utterances.

## B    TRAINING DETAILS OF THE GENERATOR ON OVERNIGHT DATASET

The final setting has 2 encoder and 2 decoder layers. The source embedding and hidden dimensions are chosen as 64 and the target embedding and hidden dimensions are chosen as 128. We set the corresponding feed forward layer hidden dimensions 4 times higher. We employed dropout at training time with probability 0.2. We used 8 attention heads for each task. We used a clipping distance of 8 for relative position representations (Shaw et al., 2018).

We used the Adam optimizer (Kingma & Ba, 2014) with $\beta_1 = 0.9$, $\beta_2 = 0.98$, and $\epsilon = $ 1e-9. We set the learning rate as 5e-5. We used the same warmup and decay strategy for learning rate as Vaswani et al. (2017), selecting the warmup steps as 4000. We trained the model for 50000 steps with batch size 32. We used a simple strategy of splitting each input utterance on spaces to generate a sequence of tokens. We mapped any token that did not occur at least 2 times in the training dataset to a special out-of-vocabulary token. Token embeddings are taken from a pre-trained BERT (Devlin et al., 2018) encoder. We freeze the pre-trained parameters of BERT for 6000 steps, and then jointly fine tune all parameters, similar to existing approaches for gradual unfreezing (Howard & Ruder, 2018). When unfreezing the pre-trained parameters, we restart the learning rate schedule.

Motivated by the best performing approach in Herzig & Berant (2017), the generator is trained using the natural utterance, logical form pairs over all training examples, i.e., jointly over all domains. Therefore, all model parameters are shared across domains and the model is trained from all examples. Since there is no explicit representation of the domain that is being decoded, to help the model learn identifying the domain given only the input, we add an artificial token at the beginning of each source sentence to specify the target domain, similar to Johnson et al. (2017) for neural machine translation.

