# OpenReview forum: "Improving Semantic Parsing with Neural Generator-Reranker Architecture"
_ICLR.cc/2020/Conference — Reject_

### Official Review · AnonReviewer3 · 2019-10-14
**Official Blind Review #3**

**Rating:** 1

**Review:**

This paper proposes a reranking architecture with a LogicForm-to-NaturalLanguage preprocessing step for semantic parsing. The authors experiment their method on three datasets and get the state of the art results.

The proposed method is natural. But using neural models to rank (or rerank) is a long-existing technique, regardless of the chosen parametrization of the reranking model. This paper chose BERT. See section-2.6 of this tutorial for more details about using neural models to rank: https://www.microsoft.com/en-us/research/uploads/prod/2017/06/INR-061-Mitra-neuralir-intro.pdf.

Overall, I think the paper is not ready to publish for the following reasons.

1. The method relies much upon manual designs that seem hard to generalize.

By converting the logic forms to natural languages, the authors can leverage paraphrase datasets and pre-train the critic as a paraphrase model. However, the way they convert the logic forms is different for each dataset and they have to manually design rules for each logic form.

2. It is not clear how certain experimental designs were made.

The authors chose to not rerank if the candidates' scores are too low or high but close. Such choice and associated thresholds seem arbitrary: how were they actually found out? Were they tuned on a development set? How does the method work if the candidate with the highest score is always picked: in the end, this is what the model is supposed to learn, correct?

Other designs include beam size, whether or not to use a pretrained model, etc. How were such decisions made? Tuned on a development set?

3. The results are not sound enough. Given the issued pointed out in 1 and 2, I am not sure if the results are really sound as the authors claimed.

For example, what if the authors don’t use a LogicForm-to-NaturalLanguage conversion? What is the result if we directly learn to match input and logic forms?

Moreover, the authors better answer questions in 2 so I can gauge if their hyper-parameters were chosen in the principled ways. Once those are answered, a significant test had better be done since the improvement seems small.

4. Claiming Shaw et al. 2019 in table-3 as ``our methods’’ is wrong. It is clear that Shaw et al. (2019) didn't experiment on OVERNIGHT dataset, but setting up the baseline on a dataset should not be classified as ``our method’’.

Moreover, I have some comments on the model and experiments. These are not weakness, but I think some work in this direction may help improve the paper.

1. The model architecture should be better justified. In its current form, the two arguments (input query and output sequence translated from a logic form) are interchangeable. Why so? Why isn’t an asymmetric architecture more natural? How can the authors use a pair of logic forms as negative examples (in figure-2)? Why do the authors use the Quora dataset in particular?

2. The error analysis might be better to be a bit more quantitative. Its current form doesn’t seem to give insight on how the proposed method really helps. What the authors can do is: you can sample some sentences from the test/development set and count how many comparative words are misused in the original model, among which how many are corrected by reranking.


**Experience Assessment:**

I have read many papers in this area.

**Review Assessment: Checking Correctness Of Derivations And Theory:**

N/A

**Review Assessment: Checking Correctness Of Experiments:**

I carefully checked the experiments.

**Review Assessment: Thoroughness In Paper Reading:**

I read the paper thoroughly.

---

> ### Author Response · Authors · 2019-11-14
> **We thank the reviewer for the evaluation of our paper and the comments. We firmly believe that the comments of the reviewer can be addressed through additional clarifications and below we provide a point by point response to the comments of the reviewer.**
>
> Response to C1: We provide 3 methods for the conversion of logical forms. We believe only the last method could be challenging to generalize to all datasets. We point out that our first method does not even convert logical forms and it uses them as they are, therefore, is trivially applicable to any dataset. Our second method simply converts entities to natural text and this would only require a dictionary specific to a dataset entity names, which we believe is easily applicable to any dataset. We emphasize that these two methods already provide strong improvements over the baseline. For instance, the first two methods improve the performance of the baseline from 82.1% to 82.81% and 83.16% respectively for the overnight dataset. Similarly, for GEO, we use the first method, which does not change the logical forms, and obtain an improvement over the baseline from 92.5% to 93.2%. For the ATIS dataset, we use the second method and obtain an improvement over the baseline from 89.7% to 91.5%. We point out that method 3, which is arguably harder to generalize to other datasets, is only used on the overnight dataset. However, the first two methods already provide strong improvements on all 3 datasets.
>
> Response to C2: We introduce two thresholding strategies. The first one is not reranking if all candidate scores are below 0.5. We note that 0.5 is the boundary value for a binary classifier, which means if all candidate scores are below 0.5, this essentially means that the critic predicts none of these candidates as correct for a given natural sentence. Therefore, it is very intuitive to choose not to rerank in this case.
>
> We agree that the second threshold strategy may look arbitrary and we stated this in the paper as well. Here, we choose not to rerank if two candidate scores are high but closer than 0.001 in difference. We did not tune this threshold value and picked it by looking at the model mistakes on the training set. Actually, tuning this on a dev set would help us even further and improve the performance. However, getting a specific threshold value (e.g. 0.0157) out of tuning over the development set would look even more arbitrary in our opinion. Therefore, we set this threshold and kept it as it is throughout all experiments we provide in this paper.
>
> In the experiments, we provide the results when the candidate with the highest score is always picked. Note that, all the results that do not include TH1 or TH2 picks the candidate with the highest score. This improves the baseline performance from 82.1% to 83.5% in the overnight dataset as an example.
>
> We note that increasing the beam size helps the model since there is more chance that the correct form appears among the candidates and can be picked by the critic. However, the performance did not increase significantly above beam size 25 so we concluded there. In terms of using a pretrained model, we found that pretraining always helps on the development set, therefore, it is our advantage to use a pretrained model in all cases.
>
> Response to C3: Regarding not using a LogicForm-to-NaturalLanguage conversion, we point out that the first method (Raw Logical Form) precisely covers this case. In this method, the logical forms are used as they are and we do not do any conversion. The critic directly learns to match input and logic forms and we provide the results of this setting in the experiments. For the overnight dataset, this improves the baseline performance from 82.1% to 82.81%. This is indeed a significant improvement because the overnight dataset consists of 8 different domains and the results reported here are the average accuracy over 8 domains. We provide a similar result for GEO dataset. Therefore, even when the critic directly learns to match input and logic forms, we get improvements over the baseline and obtain SOTA results. Additionally, error analysis shows typical mistakes the baseline makes and the critic corrects. We observe that the improvements made by the critic are significant.
>
> Response to C4: We apologize as we did not mean to make such a claim. We will fix this position in the table.
>
> Response to C5: We note that we do have an asymmetric architecture where we compare a query directly with a logical form and as explained above this setting already results in an improvement in our architecture.
>
> A pair of different logical forms basically represents different queries, therefore, this can be considered as a negative example and allows us to increase the number of training examples and also helps the critic to learn differences among them.
>
> We used the Quora dataset because it is a widely used dataset that has over 400K annotated question pairs. We have also tried other datasets such as PAWS dataset, however, we didn't see further improvement hence we continued with the Quora dataset.
>
> Response to C6: We thank the reviewer for this suggestion. We will make the error analysis more quantitative by following the suggestion of the reviewer.

---

### Official Review · AnonReviewer2 · 2019-10-21
**Official Blind Review #2**

**Rating:** 3

**Review:**

This paper proposes a framework for semantic parsing, which includes a neural generator that synthesizes the logical forms from natural language utterances, and a neural reranker that re-ranks the top predictions generated by beam search decoding using the neural generator. While the neural generator is the same as prior work, the main novelty is the reranker design, which is a binary classifier that takes a pair of natural language utterance/logical form, and predicts the similarity between them. This reranker could also be pre-trained using auxiliary data sources, e.g., Quora question pairs benchmark for paraphrasing. They evaluate their approach on 3 semantic parsing datasets (GEO, ATIS, and OVERNIGHT), and show that their reranker can further improve the performance of the base generator.

I think the general motivation of the framework is sound. Although the idea of reranking is not new in the semantic parsing community, with the most recent work [1] already shows the promise of this direction, the concrete approach described in this paper is different, seems simple yet effective. The most interesting part is to transform the generated logical form into a pseudo-natural language text, so that it becomes a paraphrase of the input natural language utterance in some sense, which enables the re-ranker to be pre-trained with auxiliary data sources, and to use the wordpiece tokenizer that is effective in understanding natural language. In their evaluation, they indeed show that this transformation helps improve the performance of the reranker.

My main concern of  this paper is about evaluation. First, although they already evaluate on 3 datasets, all of them are not among the most challenging benchmarks in semantic parsing. In [1], they also evaluate on Django and Conala, which are 2 benchmarks to translate natural language to Python, and also are more complicated than the benchmarks in this paper. It would be helpful for the authors to show results on such datasets that the results of baseline neural generators are less satisfactory, which may also make more room for the possible improvement using a re-ranker.

On the other hand, they also lack a comparison with existing re-ranking approaches. For example, it will be helpful to compare with [1], given that they also evaluate on GEO and ATIS. Right now the results are not directly comparable because: (1) the base generators are different; and (2) the beam size used in this paper (10) is larger than the beam size (5) in [1]. It will be helpful if the authors can at least provide results with a smaller beam size, and would be better if they can provide results that are directly comparable to [1].

[1] Yin and Neubig, Reranking for Neural Semantic Parsing, ACL 2019.

------------
Post-rebuttal comments

I thank the authors for the response. However, I don't think my concerns are addressed; e.g., without a comparison with previous re-ranking methods, it is hard to justify their proposed approach, given that other re-ranking methods are also able to improve over an existing well-performed generator. Therefore, I keep my original assessment.
------------

**Experience Assessment:**

I have published one or two papers in this area.

**Review Assessment: Checking Correctness Of Derivations And Theory:**

I carefully checked the derivations and theory.

**Review Assessment: Checking Correctness Of Experiments:**

I carefully checked the experiments.

**Review Assessment: Thoroughness In Paper Reading:**

I read the paper thoroughly.

---

> ### Author Response · Authors · 2019-11-14
> **We thank the reviewer for the evaluation of our paper and the comments. Below we provide a point by point response to the comments of the reviewer.**
>
> Comment: My main concern of this paper is about evaluation. First, although they already evaluate on 3 datasets, all of them are not among the most challenging benchmarks in semantic parsing. In [1], they also evaluate on Django and Conala, which are 2 benchmarks to translate natural language to Python, and also are more complicated than the benchmarks in this paper. It would be helpful for the authors to show results on such datasets that the results of baseline neural generators are less satisfactory, which may also make more room for the possible improvement using a re-ranker.
>
> Response: We agree with the reviewer that applying our architecture on more benchmark datasets would make this work stronger and we also stated this as a future work. However, we believe that significantly improving the best performing model over three widely used semantic parsing datasets and achieving the state-of-the-art results, along with showing through error analysis how our architecture is systematically helpful is already a very strong and convincing argument of the fact that the introduced reranker model is very effective.
>
> We note that the choice for the datasets we used were not arbitrary. We wanted to test with the best performing generator model because if we were to use a weaker model, then the improvement would be questionable and one could easily argue whether the introduced approach can improve the best model in the literature. Therefore, it was important for us to show that our approach can improve upon already the best performing model in the literature. We chose the overnight dataset because this is a dataset that has 8 various domains with large number of examples and allows for all the three processing methods we proposed to be experimented. We chose the GEO and ATIS datasets because the generator we use was not applied to the overnight dataset, but it was applied to these two datasets. So we wanted to show that we can also improve the performance on these datasets as well. Of course, evaluating on more benchmark datasets (like challenging ones such as Django and Conala) would make our results stronger, however, we believe our set of results are already significant and can show the effectiveness of the reranker model we introduced.
>
> Comment: On the other hand, they also lack a comparison with existing re-ranking approaches. For example, it will be helpful to compare with [1], given that they also evaluate on GEO and ATIS. Right now the results are not directly comparable because: (1) the base generators are different; and (2) the beam size used in this paper (10) is larger than the beam size (5) in [1]. It will be helpful if the authors can at least provide results with a smaller beam size, and would be better if they can provide results that are directly comparable to [1].
>
> Response: We agree with the reviewer that the results are not directly comparable and we would like to emphasize that our approach improves upon the best performing model in the literature, which is a more challenging improvement. We agree with the reviewer that a direct comparison would be better and we will do this comparison in future work.

---

### Official Review · AnonReviewer1 · 2019-10-25
**Official Blind Review #1**

**Rating:** 3

**Review:**

In this paper, a method for re-ranking beam search results for semantic parsing is introduced and experimentally evaluated. The general idea is to train a paraphrase critic. Then, the critic is applied to the each pair (input sentence, logic form) in the beam to determine if they are close.

The main problem with the proposed method is that the critic does not receive high quality negative examples. The generator is never trained to adapt to the critic. Second big problem is that the critic trained on two sources of data: the original dataset and the Quora paraphrasing dataset. It is very unclear what is the impact of each of the data sources. Also, it is unclear how the critic works in this case. It seems to be an easy task to distinguish a logical form from a natural sentence.

In general, the paper is well written. I would suggest to reduce the size of the introduction and dedicate this space to more detailed explanation how reranking works and the experimental details. Figures don't add much to understanding.

The experimental part is rather weak. The error analysis part is great, but not very methodical. It is not clear is these examples are cherry picked or it is frequent mistake of the baseline. I would like to the accuracy of the critic and the analysis of its performance. The critic is the main contribution of this paper and it is strange that so little attention is dedicated to it. Other aspects that need to be highlighted in the experimental section:
- how the Quora pretraining helps
- do other strategies for negative sample work
- how important is not to rerank in certain cases (Sec 3.3)

In conclusion, I encourage the authors to develop the idea further. Taking in into account the issues with the method (or its presentation) and the experimental weaknesses I recommend reject for now.

Typos:
- [CLS] is not defined in text
- Shaw et al. should be in Previous methods in Table 3

**Experience Assessment:**

I have read many papers in this area.

**Review Assessment: Checking Correctness Of Derivations And Theory:**

I carefully checked the derivations and theory.

**Review Assessment: Checking Correctness Of Experiments:**

I carefully checked the experiments.

**Review Assessment: Thoroughness In Paper Reading:**

I read the paper at least twice and used my best judgement in assessing the paper.

---

> ### Author Response · Authors · 2019-11-14
> **We thank the reviewer for the evaluation of our paper and the comments. We firmly believe that the comments of the reviewer can be addressed through additional clarifications and below we provide a point by point response to the comments of the reviewer.**
>
> Comment: The critic does not receive high quality negative examples.
>
> Response: We believe that the critic actually receives high quality negative examples through the generator. This is because the generator is the best performing model in the literature and for each training example, through beam search, we are able to obtain candidates that are incorrect but very similar to the gold label. Therefore, training the critic with such high quality negative examples allows the model to learn subtle differences among the candidates and make a better judgment on choosing the best fit for a given query. Hence, we believe getting negative examples through the best performing generator model is a good approach and we cannot think of a better source that can generate negative examples with better quality.
>
> Comment: The generator is never trained to adapt to the critic.
>
> Response: We agree with the reviewer that the generator and the critic are not jointly trained. In this paper, our focus was i) introducing a novel critic model; ii) arguing the advantages of this model which cannot be obtained in the generator such as observing each candidate and the input sentence entirely, taking into account bidirectional representations of both sentences and leveraging extra data sources for training; iii) showing the advantages of the critic along with how it improves the parsing performance through extensive analysis. We do believe that jointly training the generator and the critic is an interesting direction and we would like to leave this to a future work.
>
> Comment: It is very unclear what is the impact of each of the data sources.
>
> Response: We believe that pretraining the critic with extra data sources, like Quora paraphrasing dataset is one advantage of this model which cannot be realized in the generator. We point out that in the experimental setting we provide the results explicitly for each data sources. As an example, the first experiment shows that the baseline achieves 82.1% accuracy and using the critic that is trained on the original dataset improves it to 82.5%. Furthermore, if the critic is pretrained with the Quora dataset, we get 82.7%, which is a further improvement. These results are for beam size 10 and we provide similar results for beam size 25 as well. We believe this experiment clearly shows the advantage of the critic that is trained with the original dataset and also the advantage of pretraining the critic with the extra Quora dataset.
>
> Comment: Also, it is unclear how the critic works in this case. It seems to be an easy task to distinguish a logical form from a natural sentence.
>
> Response: The critic takes a pair of (natural sentence, logical form) and scores the pair based on the correspondence of the logical form to the natural sentence. This task is performed via binary classification so the output of the critic is a score between 0 and 1. A better score by critic (a score closer to 1) essentially means the logical form is more likely to be the right label for the natural sentence. This is the task of the critic and it does not change throughout the paper. This is why we get confused by the reviewer’s statements “works in this case” and “distinguish a logical form from a natural sentence” since the critic works only one way according to the description above and a logical form is not distinguished from a natural sentence, we basically score assessing whether the logical form is the correct version for the natural sentence given a (natural sentence, logical form) pair.
>
> Response to paragraph 3: We thank the reviewer for the positive comment and based on the suggestion of the reviewer, we can make the introduction shorter and explain the critic model with the reranking procedure in more detail.
>
> Response to paragraph 4: In the error analysis part, we examined the errors and created error type buckets for the mistakes of baseline model that are corrected by the critic. Therefore, these mistakes are frequent and categorized accordingly. However, we can improve this section based on the reviewer’s comment. We can sample sentences from the test set and provide quantitative results accordingly.
>
> We emphasize that the generator is the best performing model in the literature, therefore, we believe that allows us to obtain high quality negative examples to train the critic. A weaker model or a random sampling would not result in such high quality negative examples in our opinion.
>
> We believe that in the experimental section our results show the advantages of not to do reranking in certain cases. For example, for the overnight dataset, if we choose not to rerank when the critic scores all the candidates below 0.5, the performance improves from 83.5 to 83.7. We provide similar results with threshold rules (TH1, TH2, TH3) for all datasets and we believe these results show various reranking strategies one can take with the critic model.

---

> > ### Comment · AnonReviewer1 · 2019-11-14
> > **Clarification**
> >
> > > The critic takes a pair of (natural sentence, logical form) and scores the pair based on the correspondence of the logical form to the natural sentence.
> >
> > My confusion is about using the Quora dataset. What is the pair in this case? What are the positive and negative examples?

---

> > > ### Author Response · Authors · 2019-11-15
> > > **Clarification**
> > >
> > > Thank you for this clarification question.
> > >
> > > The Quora dataset consists of question pairs that are either paraphrases (label 1) or not (label 0). As an example, Question1 = "How do you start a bakery?" and Question2 = "How can one start a bakery business?" pair has the label 1 and Question1 = "What are natural numbers?" and Question2 = "What is a least natural number?" pair has the label 0.
> > >
> > > The reason for us to use this dataset to pretrain the critic model is the following. The task of the critic is to rerank the candidate logical forms based on their similarity score with respect to the input utterance. We introduce three processing methods for the logical forms that could potentially help the critic in this task. We propose the processing methods in an increasing order of their complexity
> > > and how close they map the logical forms to natural text. Namely, the first method does not change the logical forms and use them as they are; second method simply convert entities in the logical forms to natural text; third method converts logical forms to canonical utterances using a deterministic template. We point out that as the logical forms are processed closer to the natural text, the task of the critic is getting closer to the paraphrase identification task and this will be very similar to the Quora dataset examples that we described above. Therefore, pretraining our critic model on the Quora dataset and then fine tuning it over the examples we generate is an effective approach, which provides good performance.
> > >
> > > We point out that even when the logical forms are not processed (method 1, i.e. 3.2.1 Raw Logical Form), we obtain improvements over the baseline as we show in the experiments. However, processing the logical forms closer to natural text does help the critic to score their similarity with respect to the input utterance and provide a better performance.
> > >
> > > We would be happy to provide further clarifications if desired by the reviewer.

---

### Decision · Program_Chairs · 2019-12-19

**Decision:**

Reject

**Comment:**

This paper presents and evaluates a technique for semantic parsing, and in particular proposes a model to re-rank the candidates generated by beam search. The paper was reviewed by 3 experts and received Reject, Weak Reject, and Weak Reject opinions. The reviews identified strengths of the paper but also significant concerns, mostly centered around the experimental evaluation (including choice of datasets, lack of direct comparison to baselines, need for more methodical and quantitative analysis, need for additional analysis, etc.) and some questions about the design of the technical approach. The authors submitted responses that addressed some of these concerns, but indicated that additional experimentation would be needed to address all of them. In light of these reviews, we are not able to recommend acceptance at this time, but I hope authors use the detailed, constructive feedback to improve the paper for another venue.